# Developing a Cyber Incident Exercises Model to Educate Security Teams

Basil Alothman * 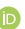, Aldanah Alhajraf, Reem Alajmi, Rawan Al Farraj, Nourah Alshareef and Murad Khan

Kuwait College of Science and Technology, Doha District 35001, Kuwait; 171032@kcst.edu.kw (A.A.);
171036@kcst.edu.kw (R.A.); 172139@kcst.edu.kw (R.A.F.); 181211@kcst.edu.kw (N.A.);
m.khan@kcst.edu.kw (M.K.)
* Correspondence: b.alothman@kcst.edu.kw

**Abstract:** Since cyber attacks are increasing and evolving rapidly, the need to enhance cyber-security defense is crucial. A cyber incident exercise model is a learning technique to provide knowledge about cyber security to enhance a security team's incident response. In this research work, we proposed a cyber incident model to handle real-time security attacks in various scenarios. The proposed model consisted of three teams: (1) the black team, (2) the red team, and (3) the blue team. The black team was a group of instructors responsible for setting up the environment. They had to educate the red and blue teams about cyber security and train them on facing cyber attacks. Once the training period was completed, the members were divided into two teams to conduct a cyber-security competition in a cyber game scenario. Each of the two teams performed a different task. The red team was the offensive team that was responsible for launching cyber-security attacks. The blue team was the defensive team that was responsible for countering attacks and minimizing the damage caused by attackers; they had to conduct both cyber-security configuration and incident handling. During the scenario, the black team was responsible for guiding and monitoring both the red and the blue teams, ensuring the rules were applied throughout the competition. At the end of the competition, the members of each team changed with each other to make sure every team member was using the knowledge they gained from the training period and every participant was evaluated impartially. Finally, we showed the security team's offensive and defensive skills via the red team and the blue team, respectively.

**Keywords:** cyber education; cyber lab; cyber classroom; cyber incident exercises; information security teams; cyber scenarios

## 1. Introduction

Educating security teams is difficult because of the lack of real-life cyber scenarios and defensive skills to stop various types of attacks. Similarly, the instructor should prepare the cyber lab environment with a real attack and defense scenario to provide skills to the cyberwar members. In addition, the instructor will make sure that the security members enhance their skills and benefit from the exercise. Similarly, the defensive and offensive team switch their roles and conduct the exercise again. Further, the security team will be able to complete a cyber-security defense and incident-handling program to improve their security skills and be ready to face any cyber attacks if they occur.

The cyber range (CR) comprises the interactive, virtual versions of a company's local network, system, tools, and applications that are linked to a simulated internet environment [1–3]. These provide a secure environment for product development and security posture assessment, as well as a safe, legal environment for gaining hands-on cyber skills. A CR could include both hardware and software, or it could be a mix of both physical and virtual components. Other CR settings may be interoperable with ranges. The internet level of the range environment comprises not only simulated traffic but also network services such as webpages, browsers, and email, which the consumer requires.

Nowadays, cyber security is at the highest point of numerous administration's plans, and a broad examination is required to plan arrangements that ensure against or moderate cyber attacks. To assess such arrangements and to build comprehension of how cyber attacks against associations advance and proliferate, the replication of sensible assault and guard situations is vital [4,5]. In this research, the main focus is to design cyber attack scenarios to enhance education, awareness, and defensive skills to deal with currently expected cyber attacks that cover all cyber incident models in a safe environment (sandbox), which protects us from hackers with real cyber-attack tools. Further, ethical hacking is defined as a legal attempt to gain unauthorized access to a computer system, an application, or data [6]. An aspect of carrying out an ethical hack involves replicating the techniques and habits of hostile attackers. This method aids in the detection of security issues, which can then be rectified before they are exploited by a malevolent attacker. Furthermore, ethical hackers, sometimes known as white hats, are security experts who perform these assessments. They contribute to the security posture of a business by being proactive. Additionally, ethical hacking differs from criminal hacking in that it requires permission from the organization or owner of the organization before proceeding. Similarly, cyber attacks are evolving the need for cyber-security defense improvements, which is why we developed a real cyber-incident exercise model that will help enhance security teams' incident response, train them in dealing with cyber-attack handling, identify their weaknesses, and increase their threat awareness [7]. Apart from the above discussion, there are a number of models that exist, with various teams presented with various colors, as presented in Table 1. However, these models have various issues and limitations, and we concluded them in Section 3.

This paper presented a solution to enhance the security teams' cyber-security offensive and defensive skills in the academic domain by facilitating real-life cyber-attack scenarios and hands-on cyber-attack and defense-learning practices. Research conducted on open-source websites has led to the conclusion that all of them have a limitation on some tools that are needed for the intended cyber range. Therefore, the building of a cyber-range site from scratch has begun to understand exactly what to expect and to have the ability to improve the site by introducing new features in the future. In the beginning, the instructors will set up the environment and have the responsibility of educating and training the security members about cyber-security attacks and incident handling. Once the training period is completed, the security members will be divided into two teams, i.e., red and blue, as shown in Figure 1, to conduct a cyber-security competition in a game scenario. Each one of the two teams will have a different task to perform. For instance, the red team is the offensive team responsible for launching cyber-security attacks. On the other hand, the blue team is the defensive team that is responsible for countering attacks and minimizing the damage caused by attackers. Similarly, they must conduct both cyber-security configuration and incident handling. The black team is the team responsible for guiding and monitoring both the red and the blue teams, making sure that the rules are applied throughout the competition. In each round of the competition, the security members will change teams to make sure every member is applying the knowledge they gained from the training period, so that everyone will be evaluated fairly. The purpose of this exercise is to detect attacks, propose countermeasures, and develop a better incident response rather than to fix vulnerabilities and launch attacks.

The remainder of this paper is divided into following sections. Section 2 presents the related studies published in the domain of cyber security. Section 3 presents a discussion of the current challenges that exist in the cyber-security domain similar to the proposed study. Section 4 outlines the proposed cyber incident exercise model. The discussion on results is presented in Section 5. Finally, the conclusion is given in Section 6.

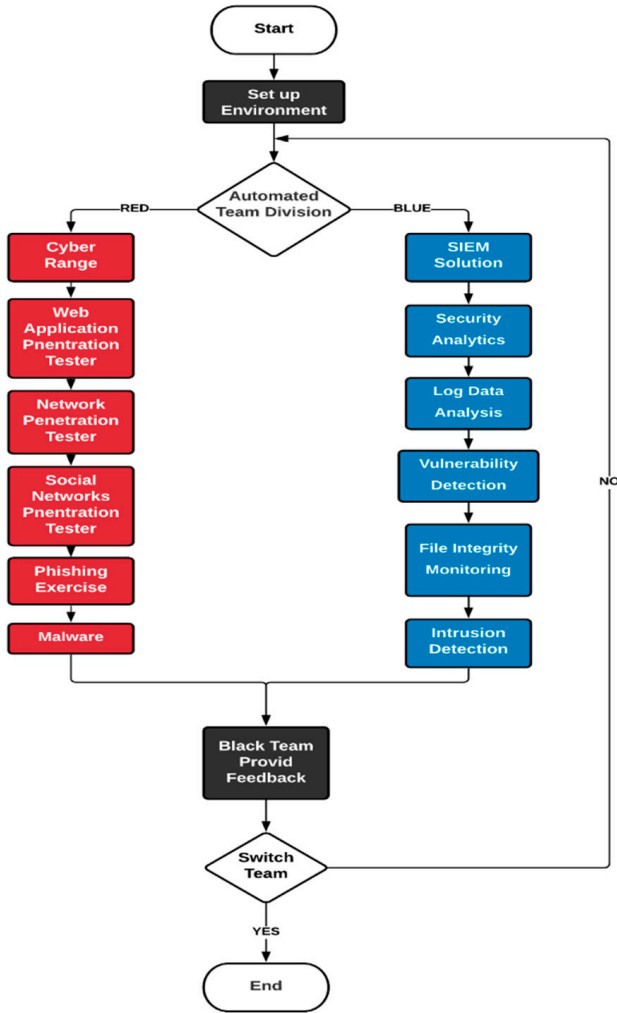

**Figure 1.** The division of teams into red and blue.

## 2. Literature Review

This research focuses on designing a cyber-attack and cyber-defense model to enhance the education and awareness skills to deal with currently expected cyber attacks that cover all cyber incident models in a safe environment. In the following paragraphs, we have concluded the most relevant research work and resources related to the proposed scheme that will help give more information about studies that revolve around cyber security, Security Information and Event Management (SIEM) solutions [8], CR, and cyber-security labs used in education.

A research work conducted in [9] deals with three main ideas: first, knowing how exercises are defined, and the current trend in this area; second, the different use of CR testbeds to simulate and test network attack scenarios such as Sonarqube [9], DETER-lab [10], ARENA [11], Emulab [12], RINSE [13], LARIAT [14], and some universities [15] that use the Moodle e-Learning system for cyber education; and third, using separate environments such as CR and SonarQube as a key tool for analysis and supervising the exercises. The proposed scenario in [9] trains people on source code analysis and a vulnerability detection approach, representing a first step in defining a CR infrastructure and a cost-effective exercise using SonarQube. The difference between the proposed system presented in [9] and our approach is that the former used SonarQube, a static source code analysis tool and the later uses the Q8CR approach. At the same time, our proposed approach is a cyber-security incident model that deals with different scenarios, including source code analysis.

The authors in [16] attempt a systematic survey of 10 CR that have been developed in the last decade. Structured interview results were presented in a guide to developing a CR for the University of West Attica (UNIWA), while the authors discussed the information and details about tools used to design, create, implement, and operate a CR platform and presented the possible results to be used in different types of CR objectives such as research, training, exercises, education, operations, testing, academic, military, government, private enterprise, industry, demonstrations, development, testing, emulation, and simulation.

Another research work presented in [17] shows the leading and vital role of CR's advance of Cyber Defense Situation Awareness (CDSA). The main feature of the proposed study shows the importance of the evolution of participants and teams in CDSA and of back connection with other CR. Similarly, it focuses on training the model under a cyber defense situation awareness simulation to mimic real-life scenarios. This research is limited to web application security systems and collaborative scenarios. The focused research was in CDSA with two scenarios: (1) a web application firewall and (2) collaborative scenarios. Unlike our study, which supports a scripted simulation of attacks, with more than 60 different scenarios, our proposed scheme focused on developing a cyber incident exercises model to educate security teams under other security domains. Further, the authors in [17] report the efficiency of a two-week training period for a cyber-security training exercise. The training was divided into two phases. The first phase was a week of classroom training focused on educating the participants about cyber defenses. The second phase of the program was a week of a red or blue team (capture the flag) exercise where they executed their training. The participants finished the training week with better security knowledge and learned new defensive methods and techniques. While this research focused on enhancing the participants' defensive abilities, our research goal will be to train students on using Q8CR to have a better response time and minimize the attack risks. Similar research to [17] is presented in [18] to determine different ways of optimizing the detection of sophisticated attacks in Security Operations Center (SOC) environments. Their interviews with the SOC analysts confirmed some of the literature review findings and provided valuable insights into other areas of malware detection, which seem to be neglected in the literature. For future research work, they plan to implement a framework that aims to categorize network traffic into either benign or malicious clusters.

Recently, several research teams have been working in the domain of CR, as shown in Table 1. The majority of these research teams use either simulation or emulation, according to the study. The ability to provide a realistic environment for training and testing, as well as the ability to undertake high-quality and repeatable tests, are the key benefits of emulation CR. Large simulations may now be run on relatively low-cost technology. The simulation CR, in other words, is highly scalable, versatile, and low-cost. Some simulations have been criticized for the fact that verifying that they truly mirror reality is difficult.

A cyber-security team in the Austrian Institute of Technology (AIT) presented the design considerations, architecture, and implementation of the CR in [18]. The CR is a flexible, scalable, and virtual environment to support exercises, training, and research. With the CR, many exercises and training courses have been successfully held. For their future work, they aim to investigate the utilization of the CR for the interoperability and federation of CR [18]. Similar research deals with three main ideas designed for performing a lightweight systematic mapping to identify research works related to CR [10]. They were interested in knowing how exercises are defined and the current trend in this area. They are proposing the use of separate environments such as a CR, and the use of SonarQube as a key tool for analysis and supervising the exercises. The proposed exercise stems from an academic environment, but it can be used for training the workforce from different companies requiring the ability to train people on source code analysis and vulnerability-detection approaches, so it represents a first step in defining a CR infrastructure and a cost-effective exercise by using SonarQube. A small tool controls the exercises. For their future work, they are working on the identification of a new exercise to be deployed and used with a CR [10].

Another research work presented in [11] showed promising results in finding cyber security and data mining to be important, and for some, a potential career path. This research work improved the undergraduate performance in computer programming, where only one student was doing a computer science major and the rest of the students was having computer engineering as major. Similarly, the future work of this study can explore the impact of their activities and projects to further improve the program's effectiveness and pique student interest in cyber security.

A group of researchers from Morehouse College USA examined and surveyed the outcomes of a 21-week program in which minority undergraduate college students, all individuals from the Reserve Officer Training Corps (ROOC), were taught computer programming, natural language processing, data visualization, and computer vision fundamentals. Students who are not specialized in computer can compensate for the lack of individuals in cyber security. To do so, it is important to expose these individuals to data mining, cyber-security practices, and the application of these concepts in the field. The results of this paper recorded participant attitudes and self-efficacy, which are all indicators of the effectiveness of the program [19].

**Table 1.** The colors used by cyber exercise teams/roles, which identified intended users.

| Cyber Security | Blue Team | Red Team | Green Team | Yellow Team | White Team | Purple Team |
|---|---|---|---|---|---|---|
| De Montfort University [20] | ✓ | ✓ | ✓ | ✓ | ✓ | ✓ |
| Royal Military Academy [21] | ✓ | | | | | |
| Masaryk University [22] | ✓ | ✓ | ✓ | ✓ | ✓ | ✓ |
| AIT [23] | ✓ | ✓ | ✓ | ✓ | ✓ | ✓ |
| Norwegian University of Science and Technology [24] | ✓ | ✓ | | | ✓ | ✓ |
| Universite degli Studi di Milano [25] | ✓ | ✓ | ✓ | | | |
| JAMK University of Applied Sciences [26] | ✓ | ✓ | ✓ | ✓ | ✓ | ✓ |
| Swedish Defense Research Agency [27] | ✓ | ✓ | ✓ | | ✓ | |

The advantages of using QEMU-IOL virtualization technology to build industrial security CR are discussed in [28]. To check the results, performance tests were implemented on the industrial security CR to verify the stability of the CR. According to this research, the range can run industrial control application software, realize the safety test of industrial control equipment, and achieve a predetermined goal. The suggested technique could successfully share in the control, calculation, storage, and networking of the industrial control network range, greatly reducing the setting and maintenance of industrial control computers and network resources in the network range. Second, the industrial control cyber-security test requires a complex and changeable topology.

In conclusion many universities, organizations, and IT firms used colors in the cyber-security field to represent the different security teams. We have identified those colors and presented them in Table 1. These models have a number of issues and challenges, which we identify in Section 3.

## 3. Issues and Challenges

From the above literature, we identified various issues and challenges of IT security teams, including the Security Operation Centre (SOC) or Computer Security Incident Response Team (CSIRT). Many times, IT security teams do not know how to handle a real-life cyber attack. Further, they face many problems when trying to react effectively to several different types of cyber attacks. For instance, some of the participants have difficulties in understanding English and need help because their mother tongue is not the English language, while some of the participants have a lack of cyber skills experience,

specifically they do not have actual real-life cyber training skills in cyber security defensive and offensive skills. Similarly, critical national infrastructures are the main targets of cyber attacks due to the fact they contain essential information, as stated in [10]. In addition, cyber-security education using cyber-security exercises utilizing a real cyber-simulation-incident-handling environment is an emerging field that faces many challenges. These challenges are broadly divided into two types. First, issues and challenges are related to cyber-security educational programs that usually apply educational plans and convey learning preparation programs explicitly intended to fulfill the demands of cyber-security personnel. Secondly, the implementation of cyber security should be carried out in real-time to educate concerned people regarding the importance of cyber security.

## 4. Materials and Methods

In this research work, we developed a Q8CR stands for Kuwait CR that runs on cyber-security attack and defense simulation, as shown in Figure 2. Once the users enter the Q8CR system platform, they must choose the type of team to login into their accounts or create an account if they do not have one. The black team is provided with tools to create scenarios and add multiple steps to each scenario. An efficient setup for Q8CR should be equipped with many real-time scenarios and handle them as per the latest cyber-security attack simulation. CR plays a role in creating trained cyber experts who can provide innovative cyber-defense ideas; however, it has more education and training requirements for developing and launching new security products. We built our own local private CR from scratch using HTML, JAVASCRIPT, CSS, and PHP. The system is designed to provide an enthusiastic learning experience, and the students can learn how the computer network can detect any unusual behaviors and eliminate the threat. The proposed project for the lab is shown in Figure 2. The main components of the Q8CR architecture implementation for the cyber-security lab are further divided into three teams. The black team is responsible for dividing the students into two teams: a red team and a blue team. The red team starts the attack scenarios. On the other hand, the blue team starts the defending scenarios by dividing their team into the following sub-teams: the threat-hunting team, the SIEM monitoring team, the incident-handling team, and the investigative team. After that, the black team provides an evaluation and rating for each participating team.

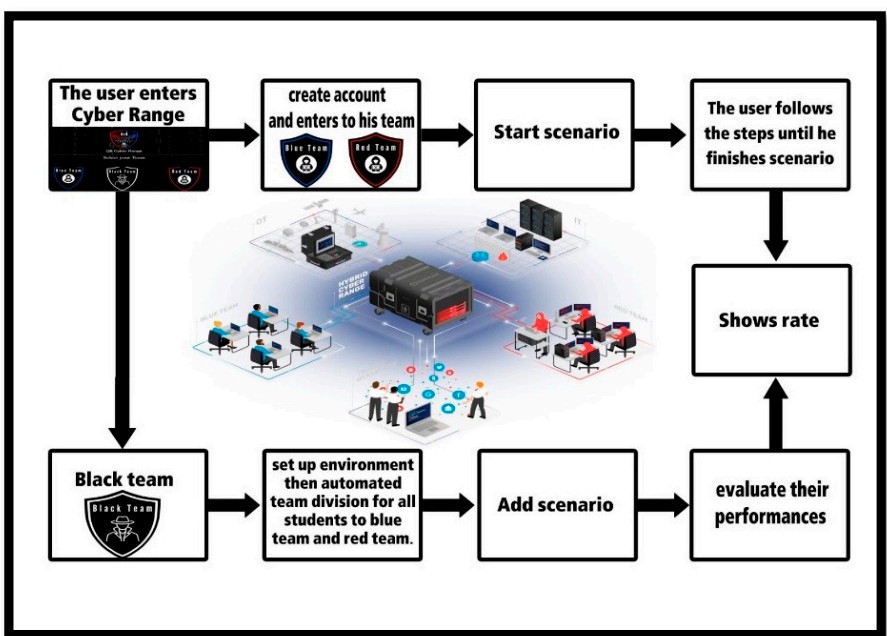

**Figure 2.** The flowchart of the proposed system.

The security team members start by logging in to the lab environment. The proposed project for the lab allows all the security team's participants to gain knowledge from both teams by swapping roles. For example, if the red team member finishes all the given scenarios mission, the student should switch to the blue team to practice. So, the black team (assessment team) starts by dividing the group into the blue team and the red team. Further, we designed a lab support room to provide information and guide both the red team and the blue team. Additionally, the Q8CR platform asks the students about their accounts, as shown in Figure 3. For instance, if the student already has an account, the request will be forwarded to the login page. However, if the student does not have an account, it moves to the main page of the platform to log in to blue team. Further, if the student already participated as a blue team member and already finished the blue scenarios, then they will join the red team and will begin by simulating the scenarios designed for the red team. Finally, the black team will evaluate the performance evaluation of each member of each team.

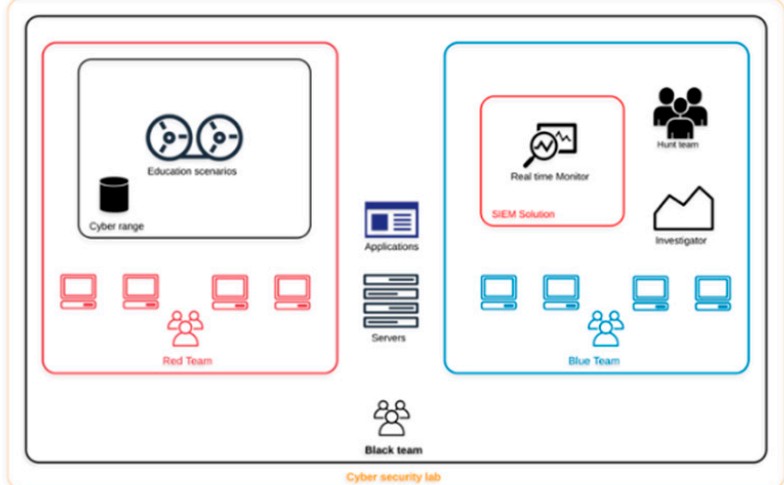

**Figure 3.** The main components of Q8CR architecture implementation.

The main login page is provided with the black team, the red team, and the blue team options. The user must choose their team to login into their accounts. However, if the user does not have an account, they have to create one and then login into their accounts. Each team is provided with their own customized login page. For instance, when the black team users log in, they are provided with multiple tools to help them create and edit the scenarios. Further, the black team has the permission to add, edit, or delete cyber-security scenarios.

In addition, they can also add users and set teams, and at the end of the scenario, they must evaluate the red and blue teams. Similarly, the red team members are provided with functionalities for launching cyber attacks on the blue team. On the other hand, the blue team is added with the functionalities of performing in the defensive mode. For instance, they must perform all the cyber-security configuration and incident handling. The red and blue teams must submit the results of their scenarios once they complete them. Finally, the red and blue teams switch sides to perform in different scenarios, while the red and blue teams' users can start their scenarios once they log in. They must follow the scenarios' steps and end the scenarios by uploading their results. The black team will receive their result and evaluate their performances. The black team is responsible to show the performance of the red and blue teams at the end. In addition to the test the scenarios, the red and blue teams must start their scenarios once they login into their account. Similarly, they must perform every step with a time limit to finish the scenarios. Finally, each team's ratings will be shown to each member of the blue and red teams. The proposed cyber incident exercises model has been designed using three cyber education models, i.e., red, blue, and black, to help students and instructors. Further, in the current research, interested readers

are referred to similar research presented in Table 1, where the authors present the cyber education model with other teams such as the purple, yellow, green, and white teams. However, the authors of this work divided the models into three teams (the black, red, and blue teams), and the practical implementation of the lab for the proposed work is shown in Figure 4.

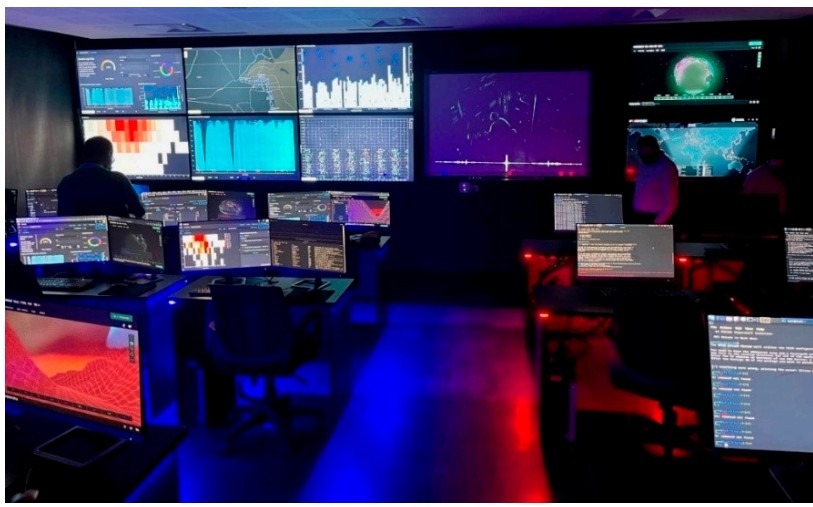

**Figure 4.** The cyber-security lab developed for practically implementing the project.

### 4.1. Black Team

The black team is responsible for creating and evaluating the scenario cases for both the red and blue teams. Similarly, each black team member should have skills and knowledge about cyber security to create step-by-step scenarios for the offensive and defensive teams to learn. They will teach and train the students about actual cyber-security attacks and incident handling. The black team has the authority to add and remove any users, create offensive and defensive scenarios, manage any technical issues, and evaluate and rate the scenario assessments. In addition, the black team is responsible for educating, communicating, and evaluating each student's performance evaluation outcome by firstly educating using the course material preparation, for instance, by showing the important rules and regulations via slide show, coursebook guidelines, and the setup of the requested lab environment. The following are some of the services that will be provided to the black team to set up the guidelines and other related activities for initiating the project.

1. User menu: user-specific information such as a user profile and a logout button.
2. Navigation: the menu includes three options, each with links to individual pages, such as users, scenarios, rating, and uploading time.
3. User types: the black team members can choose the type of team for each user.
4. Username: the member username will appear in this column.
5. Last scenario rate: the user's latest scenario rate will appear in this column.
6. Options: there will be two options in this column that will allow the black team member to access some data about the other members. The first one will be "all last group". For instance, when this option is selected, the page will open containing each user and their type of team in previous groups. Similarly, a rate button will be made available on this page that will take the black team member to a page that will show users' former ratings and scenarios.
7. Save types: the black team can transfer the users from the blue to the red team if an error occurs using the save types option.
8. Random: the random button allows the user to distribute users to the blue and red teams randomly. On the user's page, the black team user will have the ability to add users and separate them into two teams.

### 4.2. Red Team

The red team is the offensive team responsible for launching cyber attacks and penetration testing on the server side and allowing the blue team to defend their system. The login page required for a red team member is shown in Figure 5. Additionally, the red team is responsible for performing penetration testing exercises such as phishing exercises; launching malware attacks, web applications, networks; and mobile application penetration-testing using different tools installed in the lab Kali Linux operating system, as shown in Figure 4.

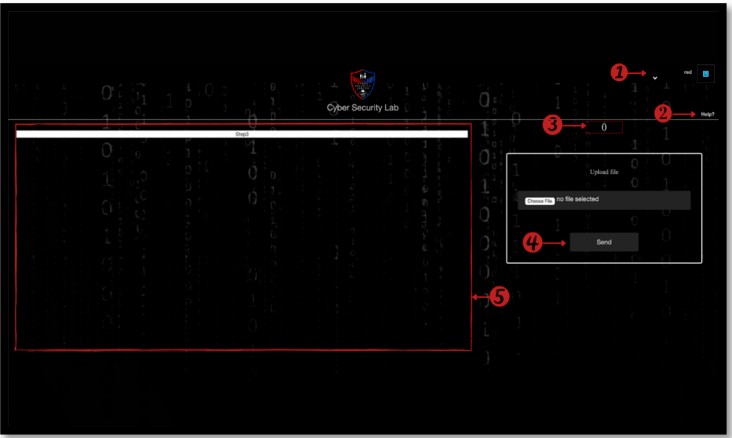

**Figure 5.** An example of the red team's login page.

1.  User menu: this contains information particular to the user, such as a profile and a logout button.
2.  Help button: this is a raised-hand button for users who want any help or to ask any questions during the exercise. They may click the help button to contact the black team.
3.  Timer: the timer displays the amount of time taken by the user to execute this step; timing is important for both teams to improve their response time.
4.  Send: the user must follow the instructions issued by the black team and upload the file for this step before proceeding to the next step.
5.  This section will display the black team member's instructions that the user must follow in this step.

As the blue team is the defensive team, they must perform all the cyber-security configuration and incident handling to counter the attacks and minimize the damage caused by the attackers. They were dealing with vulnerability assessment, threat hunting, IT security auditing, and Indicators of Compromise (IOC). The blue team can perform defense scenarios created by the black team. They start from security configuration with the ability to monitor the security logs (using the installed SIEM solution in the cyber-security lab). Further, they have threat-hunting skills and vulnerability-assessment skills provided by the step-by-step scenario. The blue team's login page is shown in Figure 6.

Based on our research goal to educate different types of cyber-security teams, we categorized them into four attack-category domains to make it easier for both the blue and red teams. Further, to facilitate the black team, they were provided with functionalities to generate different types of scenarios. Similarly, the red team was assisted to ensure the easy implementation of the various types of attacks. In this research, we have created more than 60 scenarios, which are further categorized them into two types. The first type is offensive scenarios created for the red team. These scenarios include cyber-security attack scenarios, which are categorized in Table 2. Further, we combine all the attacks under four domains. In contrast, these four domains are main categories, and the lists of attacks can be addable and updateable with the latest attack methods. We also categorize the

proposed cyber-security attack categories under the four domains, i.e., online frauds, web defacements, network attacks, and malware.

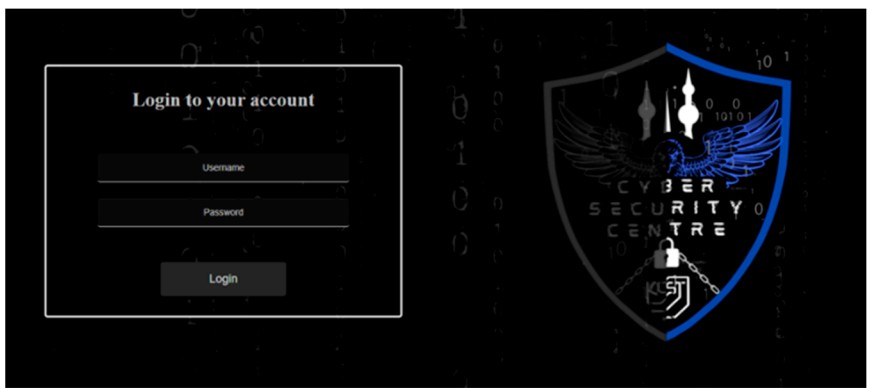

**Figure 6.** An example of the blue team's login page.

**Table 2.** The proposed cyber-security attack categories domain.

| Online Frauds | Web Defacement | Network Attack | Malware |
|---|---|---|---|
| Frauds:<br>• Online Banking Fraud<br>• Affiliate Fraud<br>• Clean Fraud<br>• Affinity Fraud<br>• Online Trading Fraud:<br>   ○ Investment Fraud<br>   ○ Shopping Fraud<br>   ○ Auction Fraud<br>• Lottery/Withdrawal Fraud<br>• Job Fraud | Injection Flaws | DDoS | Virus |
| | | Insider Threats | |
| | Broken Authentication | Login Credentials | Ransomware |
| | | Password Attack | |
| | Sensitive Data Exposure | Man-in-the-Middle (MitM) | Spy-ware |
| | | Packet Sniffer | Adware |
| | XML External Entities (XXE) | IP Spoofing | Rootkits |
| | | Flood Attack | Screen-Locking Ransomware |
| | Broken Access Control | DNS Spoofing | Trojan Horses |
| Phishing:<br>• Spear-Phishing<br>• Email-Phishing<br>• Clone-Phishing | Security Misconfiguration | Session Hijacking | Keylogger |
| | | Brute Force Attack | Drive by Download |
| | Cross-Site Scripting XSS | SSL Hijacking | Remote Access Trojan (RAT) |
| Scam | Insecure Deserialization | Compromised-Key | Riskware |
| Spam | | | |
| Identity Theft | | | |
| Social Engineering | | | |
| Vishing | Well Known Vulnerabilities | DNS Tunneling | Malvertising |
| Smishing | Insufficient Logging & Monitoring | IDS | Logic Bomb |
| | | IPS | |
| Spoofing (Email, Voice, ID) | Database Attack | Firewall Bridging | Worm |
| Pyramid Scheme | | SMTP Attack | Botnet |

The second type is defensive scenarios created for the blue team. These scenarios perform defensive techniques such as vulnerability detection, security analytics, and file-integrity detection.

## 5. Results and Discussions

The need for a cyber-incident exercises model to enhance cyber-security defensive and offensive skills entails more technical hands-on practice to avoid the lack of difficulty in acting during real cyber attacks towards a secure, robust, and stable information technology

(IT) security infrastructure. Universities can use the Q8CR model to start educating and training students using different cyber-security simulations. They can do it by including these scenarios and by avoiding the traditional cyber-security learning programs, which are unreliable to make cyber security a more appealing and accessible subject to a wide range of experts. Traditional cyber-security education and training programs usually apply conventional teaching techniques (e.g., CTF (Capture The Flag), workshops, lab meetings, group discussions, and self-learning). Second, using scenarios such as game-based methodologies can improve cyber-security learning and preparation adequacy.

In the beginning, we tested the accuracy of the attacks launched by the red team. The rate of the red team is increased with the level and severity of the attack. For instance, if the attack launched by the red team is not handled by the blue team, the rate of the red team increases and vice versa. Similarly, the blue team will respond by blocking the attack within a time limit. For instance, if the attack launched by the red team is not stopped within a given time, the rate of the blue team is minimized. For launching an attack, the red team member first selects the IP and port number of the victim machine, as shown in Figure 7a,b, respectively.

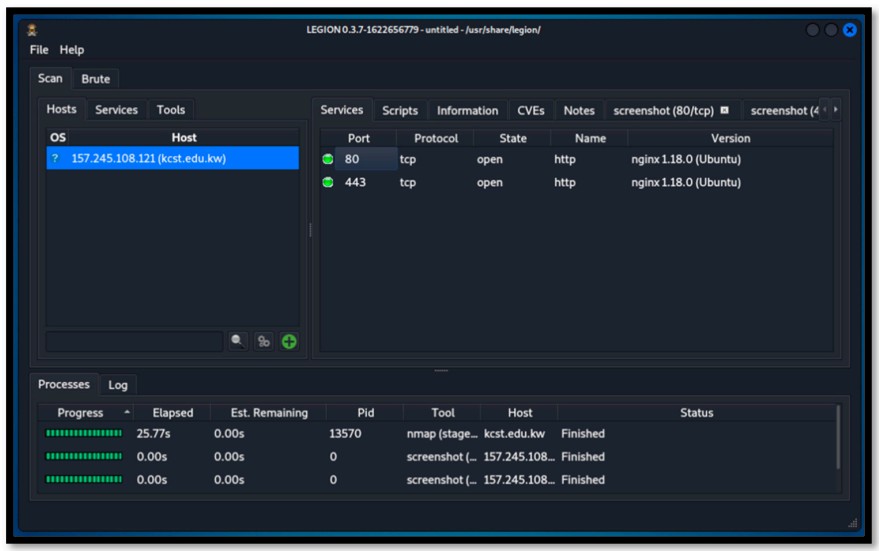

(**a**)

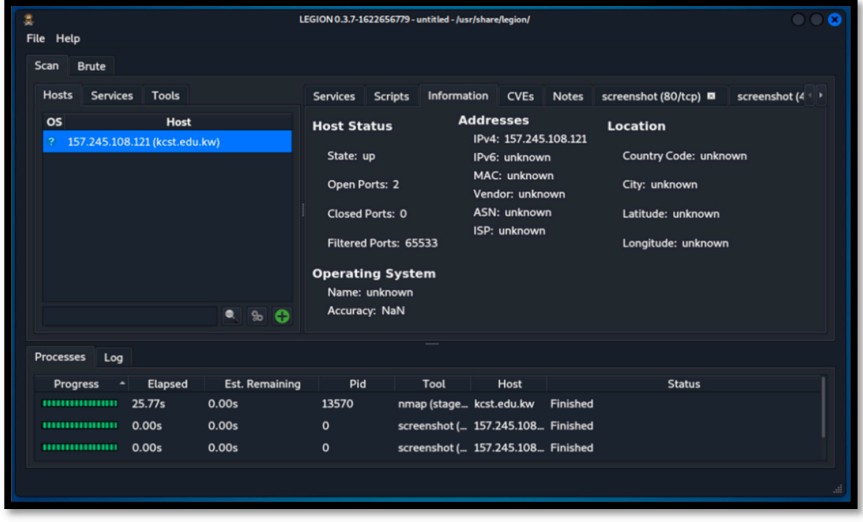

(**b**)

**Figure 7.** (**a**) Selecting the victim; (**b**) the IP address and port number.

After selecting the IP address and port number of the victim machine, the red team member launches an attack by selecting it from the various attacks available in Table 2. For instance, we have shown, in Table 2, several attacks that could be selected by the red team to launch its attack on the blue team. In response, the blue team will depend on those attacks by performing necessary countermeasures. Finally, in Figure 8, other team users' performances are presented to the black team evaluator. The black team user chooses one at a time and evaluates their step-by-step scenarios. After assessing every user, the calculated average decides which team received the best rating, ranking the accomplished solving scenarios by users to increase the level of competition.

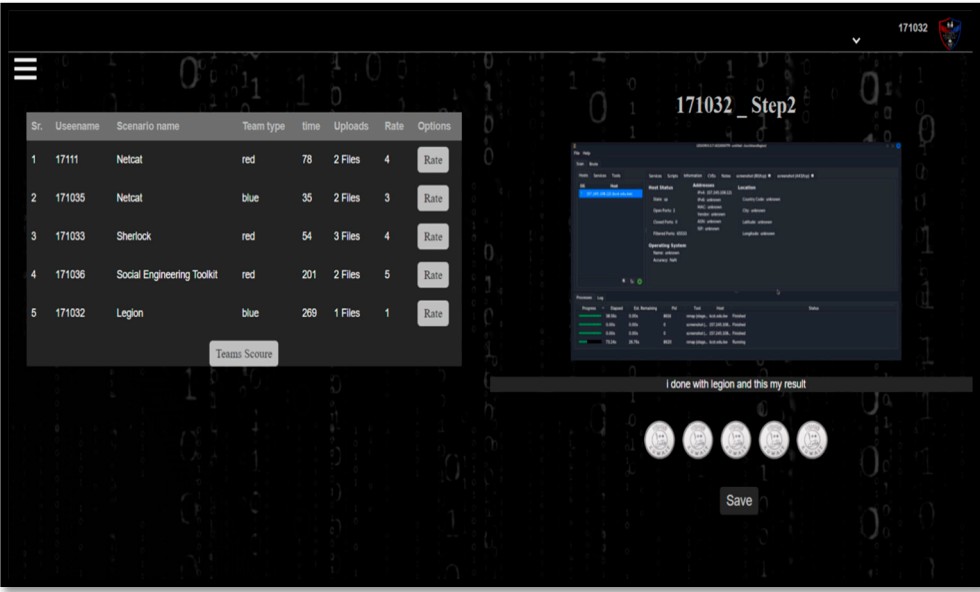

**Figure 8.** The evaluation results of the red and blue teams.

The Q8CR platform and Kali Linux tools are used to perform a real-life cyber-attack scenario to enhance the attack detection and incident handling. Additionally, they help to enhance students' knowledge about cyber attacks and help students to learn how to protect their devices and data from cyber attacks. The students get more clarity when they practice in real-life situation. This model is a complete cyber-security defense and incident handling program to improve security teams facing cyber attacks. Our proposed cyber-security educational method delivers the best possible outcome by providing the necessary knowledge and training.

## 6. Conclusions

This research work presents a new learning and effective teaching method that provides students with training and education to deal with real hands-on cyber-security attacks and incident handling. Since cyber attacks are evolving, the need for cyber-security defense improvement is crucial. We developed a real-life cyber incident exercise that educates the security teams to respond to any cyber incident that shows up. In the proposed scheme, we discussed all the requirements to accomplish the different levels and the domain type of the cyber-security exercises. The first requirement is having three teams, i.e., red, blue, and black, with different tasks to perform. The black team should prepare different cyber scenarios and answer, educate, monitor, and evaluate the participants. The red team should finalize all the given tasks of performing offensive scenarios. Finally, the blue team should have the ability to react against any offensive attacks. The second requirement is preparing the lab to educate and train the students and individuals interested in understanding cyber-security education. In order to make sure the students enhance their skills and benefit from the exercise, the blue team and the red team will switch their roles and conduct the exercise

again. Thus, we have a complete cyber-security defense and incident handling program that will improve security teams facing cyber attacks. In future work, the final requirement is to maintain and intensify the scenarios depending on the latest cyberattacks situation in the Q8CR platform to reinforce education intent, answering the common questions regarding the steps of the scenario.

**Author Contributions:** B.A. and M.K. conducted the research into the academic landscape and drafted and supervised the research. R.A.F. conducted the flowchart implementation design, while A.A. and N.A. carried out the cyber incident scenarios. N.A. and R.A. worked on the implementation, and B.A. and M.K. prepared the initial draft of the paper and evaluated the education model, while the paper was written jointly by all the authors. All authors have read and agreed to the published version of the manuscript.

**Funding:** This research received no external funding.

**Acknowledgments:** We sincerely acknowledge the Kuwait College of Science and Technology for supporting and providing a research environment to conduct this study.

**Conflicts of Interest:** The authors declare no conflict of interest.

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
