# Peer review of "Developing a Cyber Incident Exercises Model to Educate Security Teams"

_electronics, doi:10.3390/electronics11101575_

Round 1

Reviewer 1 Report

In this research work, authors proposed a cyber incident model to handle the real-time security attacks in various scenarios. However, I believe that revision is necessary to ensure the Journal's high standard of publication.  1. This research focuses on designing a cyber-attack and cyber-defense model to enhance the education and awareness skills to deal with currently expected cyber-attacks that cover all cyber incident models in a safe environment. Authors should also discuss limitations of previous methods and suggest the alternate or, best model for users.  2. Literature review is limited with only 20 references in which some of the refernces are outdated. Authors should review 10-15 more recent and relevant works and compare the proposed model.  3. Discuss part is limited especially “Evaluation results of Red and Blue team” discussion part.  4. Authors should add a table and compare the proposed model with existing one to showcase the novelty of proposed work. 

Reviewer 2 Report

This article describes the methodology followed to establish a new form of cyber security training.

First, I would like to acknowledge the work of the authors, but there are a few mistakes that should be corrected.

  • All the acronyms should be defined in the first usage and then the complete words should be avoided. The following acronyms are used without having been previously defined: CR (line 44), IT (line 67), SIEM (line 105), SOC (line 143) and AIT (line 159)
  • In line 149 CRs is mentioned and acronimed to (CR). But it makes no sense, because it is not an acronym and in the next line CRs is used
  • There are a number of errors in reference to figures:
    • line 320 refers to figure 5 as "the blue team's login page", and in the caption it is cited as "red team".
    • Line 358: figure 4 is quoted when figure 6 should be indicated. Also, the caption of figure 6 is unclear, as it is repeated twice (a)
    • Line 365: figure 4 is quoted when figure 7 should be indicated

Secondly, cyber security is compared to a soldier defending his homeland, without putting the scope in context in the introduction. It may be suitable for a conference, as it captures the attention of the audience, but I don't think it is suitable for a journal paper in its current form.

Also, in section 2 it is stated that a literature review is to be carried out. Therefore, it is to be expected that the paragraph on line 149 will cite the articles in the domain of CRs, not just mention that “many articles have been published”. It would be advisable to carry out a thorough analysis of the state of the art. The authors must analyse the published results and clearly identify the research gap to be covered. The novelty and main contributions should be a consequence of this analysis. Similarly, line 159 states that "this paper presented design considerations...", but no paper is cited. If it is the paper [18], it should be written in a clearer way, with a logical thread, which allows the state of the art to be understood. Therefore, this paragraph should be revised in depth, as it gives rise to many doubts, since later, in line 160, "a similar research" is mentioned but not cited. Mainly, this section presents a low quality analysis to become part of a research article in the selected journal.

On the other hand, in section 3 (line 187), it is stated that "These challenges are broadly divided into two types", but only one of them is mentioned. It would be useful to mention the second of these.

In section 4 (line 195), it says "as shown in our lab on a local server in KCST". I believe that most readers are not familiar with this laboratory, nor do they have access to it, so it would be interesting to modify this sentence to "which is available in the laboratory on a local server", or to refer to figure 3.

Line 253 states that "The proposed cyber incident exercises model has been designed under three cyber education models to help students and instructors". What are these three models? Similarly, the reader is then referred to other similar research in line 255, but at no point is the research cited.

The conclusions are sparse, as there is no comment on what problems or drawbacks this new method of team training may have. Furthermore, it is mentioned that "trains the security teams to respond to any cyber incident that shows up", but this is not detailed in any section of the paper, as only a series of defensive measures that the blue team would employ is mentioned above in line 338.

Therefore, with all of the above, I believe that this paper may be suitable for a conference with a little workaround, but not for publication in this journal, since in my opinion it has serious flaws. Not because of the work but due to the great work needed to improve the structure, make more experiences, analyse the state of the art properly, state the applications, results and limitations... If we take a look at the references, we find, on the one hand, that they are scarce, and on the other hand, those used have a brief, didactic content and most of them have been published in conferences. I encourage the authors to check if they would prefer to make the article suitable for a conference as well.

Round 2

Reviewer 1 Report

Satisfactory revision

Reviewer 2 Report

After analysing the authors' new version, it is worth noting the effort made, as it significantly improves the errors of the previous version. Even so, the following minor mistakes should be taken into account:

  • In figure 6 as "the red team login page", when the blue team is shown in the picture.
  • In line 396 the sentence is incomplete: "For instance, we have shown a".

In addition, the wording of the document has been significantly improved, allowing for a better understanding of the current state of cybersecurity and a comparison with existing studies.
